# The Effects of Obesity on Lymphatic Pain and Swelling in Breast Cancer Patients

**DOI:** 10.3390/biomedicines9070818

**Published:** 2021-07-14

**Authors:** Mei Rosemary Fu, Deborah Axelrod, Amber Guth, Melissa L. McTernan, Jeanna M. Qiu, Zhuzhu Zhou, Eunjung Ko, Cherlie Magny-Normilus, Joan Scagliola, Yao Wang

**Affiliations:** 1School of Nursing–Camden, Rutgers, The State University of New Jersey, Camden, NJ 08102, USA; 2Department of Surgery, School of Medicine, New York University, New York, NY 10016, USA; Deborah.Axelrod@nyulangone.org (D.A.); Amber.Guth@nyulangone.org (A.G.); 3Boston College Research Services, Chestnut Hill, MA 02467, USA; mcternam@bc.edu (M.L.M.); zhouzp@bc.edu (Z.Z.); 4Harvard Medical School, Boston, MA 02115, USA; jeanna_qiu@hms.harvard.edu; 5The Ohio State University College of Nursing, Columbus, OH 43210, USA; ko.363@buckeyemail.osu.edu; 6Boston College William F. Connell School of Nursing, Chestnut Hill, MA 02467, USA; cherlie.magny-normilus@bc.edu; 7NYU Laura and Isaac Perlmutter Cancer Center, New York, NY 10016, USA; Joan.Scagliola@nyulangone.org; 8Tandon School of Enginereeng, Electrical and Computer Engeenerng and Biomedical Engineering, New York Universuty, New York, NY 11202, USA; yaowang@nyu.edu

**Keywords:** obesity, lymphatic, pain, swelling, arm swelling, truncal swelling, lymphedema, breast cancer

## Abstract

Lymphatic pain and swelling due to lymph fluid accumulation are the most common and debilitating long-term adverse effects of cancer treatment. This study aimed to quantify the effects of obesity on lymphatic pain, arm, and truncal swelling. **Methods:** A sample of 554 breast cancer patients were enrolled in the study. Body mass index (BMI), body fat percentage, and body fat mass were measured using a bioimpedance device. Obesity was defined as a BMI ≥ 30 kg/m^2^. *The Breast Cancer and Lymphedema Symptom Experience Index* was used to measure lymphatic pain, arm, and truncal swelling. Multivariable logistic regression models were used to estimate the odds ratio (OR) with 95% confidence interval (CI) to quantify the effects of obesity. **Results:** Controlling for clinical and demographic characteristics as well as body fat percentage, obesity had the greatest effects on lymphatic pain (OR 3.49, 95% CI 1.87–6.50; *p* < 0.001) and arm swelling (OR 3.98, 95% CI 1.82–4.43; *p* < 0.001). **Conclusions:** Obesity is a significant risk factor for lymphatic pain and arm swelling in breast cancer patients. Obesity, lymphatic pain, and swelling are inflammatory conditions. Future study should explore the inflammatory pathways and understand the molecular mechanisms to find a cure.

## 1. Introduction

Obesity is the accumulation of body fat and an inflammatory condition [1,2,3,4]. The standard criterion to define obesity is a body mass index (BMI) ≥ 30 kg/m^2^ [1,3]. Obesity not only increases the risk of diagnosis and poor prognosis of breast cancer [5,6,7] but also the risk of developing long-term and debilitating adverse effects of cancer treatment [8,9]. Lymphatic pain and swelling due to lymph fluid accumulation are the most common and debilitating long-term adverse effects of cancer treatment that negatively impact breast cancer patients’ quality of life [10,11]. Over 90% of women treated for breast cancer have achieved five-year survival due to advances in cancer diagnosis and treatment [12,13]. Currently, more than 3.8 million women treated for breast cancer are living in the United States and more than 7.8 million worldwide [12,13]. Yet, there are more lost disability-adjusted life years by women to breast cancer globally than any other types of cancer [12]. The long-term and debilitating adverse effects of cancer treatment, such as pain, swelling and lymphedema, may be the major causes for the most lost disability-adjusted life years.

Patient-centered health outcomes are defined by the Patient-Centered Outcomes Research Institute (PCORI) as health outcomes that people notice and care about, such as symptoms, function, quality of life, and survival [14]. Patient-centered outcomes reflect patients’ experience in disease management and are critical indicators that help patients and healthcare providers to make ongoing value-based treatment and care decisions. Lymphatic pain refers to a variety of pain sensations (i.e., pain/aching/soreness) accompanied by swelling in the ipsilateral body or upper limb due to a compromised lymphatic system following cancer treatment [15]. Lymphedema is a chronic and incurable condition caused by an abnormal accumulation of lymph fluid in the ipsilateral body or upper limb after cancer treatment [16]. Swelling and lymphedema reflect an inflammatory pathological condition [17,18]. Swelling refers to patient-reported symptom, such as swelling in the ipsilateral upper limb (i.e., arm swelling) or upper body of breast or chest wall (i.e., truncal swelling), while lymphedema is often defined as an increased limb size or girth comparing affected (or lymphedematous) and unaffected limbs [19]. As the cardinal symptoms of early-stage lymphedema, lymphatic pain and swelling (i.e., arm swelling and truncal swelling) are symptoms that often precede changes in limb size or girth or a lymphedema diagnosis [18]. Without timely intervention, this early disease stage can progress into lymphedema that no surgical or medical interventions can cure [16]. In addition, arm swelling defined by an objective measure of interlimb volume or circumference differences) and lymphedema severity (defined by Common Toxicity Criteria) were less correlated with quality of life than arm symptoms (e.g., arm pain and swelling) [19]. Therefore, lymphatic pain and swelling are patient-centered health outcomes that are critical for monitoring the risk of and treating cancer-related lymphedema [20].

Obesity, defined as a BMI ≥ 30 kg/m^2^, is a known risk factor for arm swelling or lymphedema defined as an increased limb size or girth comparing affected (or lymphedematous) and unaffected limbs [21,22]. However, it is unknown whether obesity is associated with patient-centered health outcomes of lymphatic pain, arm swelling, and truncal swelling. While some studies [23,24] found that breast cancer patients with obesity were more likely to develop chronic bodily pain than those with a normal weight, others found no significant differences in the level of bodily pain between breast cancer patients with obesity and those with normal weight [21]. No studies have investigated the effects of obesity on patient-centered outcomes of lymphatic pain, arm swelling and truncal swelling in breast cancer patients. Therefore, the purpose of this study was to quantify the effects of obesity defined as a BMI ≥ 30 kg/m^2^ on lymphatic pain, arm swelling, and truncal swelling in breast cancer patients. We hypothesized that breast cancer patients with a BMI ≥ 30 kg/m^2^ would have higher odds of having lymphatic pain, arm swelling, and truncal swelling than those with a BMI < 30 kg/m^2^ controlling for clinical and demographic characteristics.

## 2. Materials and Methods

### 2.1. Study Design

A cross-sectional and observational design was used.

### 2.2. Ethical Considerations

This study (IRB # 16-01665) was approved by the Institutional Review Board of NYU Langone Health in New York City of the United States. All the participants signed the written informed consent.

### 2.3. Setting

The study was conducted in the breast cancer clinic of NYU Perlmutter Cancer Center, a National Cancer Institute designated cancer center in New York City of the United States.

### 2.4. Study Population

The study population was women who were older than 21 years of age, had completed acute treatment (i.e., surgery, radiotherapy, chemotherapy) for breast cancer greater than three months before enrolling in the study, and had no signs of metastatic disease, recurrence, or other bulk diseases. We recruited 570 breast cancer survivors between December 2016 and March 2020. Patients were excluded from the study if data regarding BMI, body fat percentage, and body fat mass were incomplete. A final sample of 554 patients was included in this study. Figure 1 presents Patient Flowchart.

### 2.5. Variables and Measures

#### 2.5.1. Obesity

Anthropometric measurements were used to assess height, BMI, body fat percentage, and body fat mass. Height was measured without shoes to the nearest 0.1 cm using a digital stadiometer (Seca Corporation, Chino, CA, USA). A stand-on bioimpedance analysis (BIA) device (InBody 520, Biospace Co., Ltd., Cerritos, CA, USA) was used to measure weight without shoes to the nearest 0.05 kg and the device automatically calculated BMI (kilogram/meters squared, kg/m^2^). Obesity was defined as a BMI ≥ 30 kg/m^2^ [1]. The Inbody 520 multifrequency BIA device was also used to measure body fat percentage, and body fat mass using impedance at 5, 50, and 500 kHz.

#### 2.5.2. Lymphatic Pain, Arm Swelling, and Truncal Swelling

The Breast Cancer and Lymphedema Symptom Experience Index (BCLE-SEI) Part I, a reliable and valid self-report instrument, was used to assess self-reported lymphatic pain, arm swelling, and truncal swelling [25,26]. Adequate internal consistency was demonstrated with a Cronbach’s alpha of 0.92 for symptom occurrence. The discriminant validity of the instrument was supported by a significant difference in symptom occurrence (z = −6.938, *p* < 0.001) between breast cancer patients with and without lymphedema. A response frame of the past three months was used to ensure that the symptoms were persistent and chronic. Each item was rated on a 5-point Likert scale (i.e., 0 = no presence of a given symptom to 4 = greatest severity of a given symptom). Arm swelling is defined and operationalized as the report of swelling in the affected ipsilateral upper limb. Lymphatic pain was defined as the co-occurrence of pain and swelling in the affected ipsilateral upper limb following breast cancer treatment. We operationalized lymphatic pain as the report of co-occurring pain sensations (i.e., pain, aching, or soreness) and arm swelling. Truncal swelling was defined as swelling in the upper body and operationalized as the report of co-occurring swelling in breast(s) and chest walls.

#### 2.5.3. Demographic and Clinical Data

Demographic data were collected to include age, education, marital status, employment status, and ethnicity. Financial status was assessed by asking patients to respond to the questions regarding comfortable finance (i.e., having more than enough to make ends meet), adequate finance (i.e., having enough to make ends meet), and financial hardship (i.e., not having enough to make ends meet). Medical records were reviewed to obtain information on breast cancer diagnosis, stage of the disease, cancer location, types of surgery, lymph node procedures, types of adjuvant therapy (e.g., radiotherapy, chemotherapy, hormonal therapy), diabetes, and lymphedema diagnosis.

### 2.6. Study Procedures and Data Collection

All measures were completed during a single visit to NYU Perlmutter Cancer Center. To ensure accurate measures using the Inbody 520 multifrequency BIA device, patients were instructed to stay hydrated; not participate in vigorous weightlifting, aerobic exercise, or hot yoga; not use a sauna; and not consume alcohol during the 24 h prior to the scheduled research appointment. On the day of the visit, patients were instructed not to wear body oils, lotions, or jewelry. Patients were also instructed to limit exercise to leisure walking, consume no caffeine or food (water was encouraged), and to remove any compression garments two hours prior to the appointment. All the self-reported questionnaires were administered to the patients during the visit via a study iPad connected to the study-specific electronic database capture system.

### 2.7. Data Analysis

Data were analyzed using Stata 16 SE. Descriptive statistics were calculated for demographic and clinical characteristics. Continuous variables were summarized in terms of means, standard deviations (SD), and ranges. Categorical variables were summarized using frequencies and proportions. Confidence intervals (CI) were reported to quantify the precision of estimates.

Patients were classified into obesity (BMI ≥ 30 kg/m^2^) and non-obesity (BMI < 30 kg/m^2^) groups. Group differences in demographic and clinical characteristics were estimated using *t*-test for continuous and Pearson χ^2^ test for categorical data. Effect sizes were calculated for differences between groups (i.e., Cohen’s *d* for continuous variables and Cramer’s *V* for categorical variables).

Multivariable logistic regression models were used to estimate the odds ratio (OR) with 95% confidence interval (CI) to quantify the effects of obesity on lymphatic pain, arm swelling, and truncal swelling controlling for clinical and demographic characteristics as well as body fat percentage. Clinical characteristics selected in the regression analyses were based on known risk factors for lymphedema and chronic cancer pain [17,18,19,20,21,22,23,24,25]. These variables included types of cancer surgery (mastectomy versus lumpectomy), types of lymph node procedures (sentinel lymph node biopsy versus axillary lymph node dissection), number of lymph nodes removed, receipt of radiotherapy, receipt of chemotherapy, receipt of hormonal therapy, years elapsed since breast cancer treatment, history of diabetes, and lymphedema diagnosis. Demographic characteristics were also included in the regression analysis. Body fat percentage was included in the regression models since body fat percentage and body fat mass are considered possible indicators for obesity [3]. In addition, we were interested in the effects of obesity (BMI ≥ 30 kg/m^2^) after controlling for body fat in relation to muscle mass (i.e., body fat percentage). Collinearity between obesity (BMI ≥ 30 kg/m^2^) and body fat percentage were evaluated and had acceptable low levels of variance inflation (*VIF_Obesity_ =* 1.91; *VIF_Body fat percentage_ =* 1.92). Body fat mass was not included in the regression models due to strong collinearity between body fat percentage and body fat mass as evidenced by the variance inflation (*VIF_Body fat percentage_ =* 9.87; *VIF_Body fat mass_ =* 7.16). All the tests were conducted at 0.05 alpha level and 95% confidence interval (CI). In addition to reporting the significance of parameters, model fitness was evaluated by McFadden’s pseudo R^2^ and a link-test for the logistic regression models. We present the regression parameters in exponentiated form, which are interpreted as odds ratios. Odds ratios are a measure of effect size, with an odds ratio equal to exactly 1 indicating no effect [27]. Subsequent regression models were used to examine the specific associations between BMI and the three outcomes among patients who were obese (BMI ≥ 30 kg/m^2^), using scatterplots to inform the possible linear or polynomial trends to be tested.

## 3. Results

### 3.1. Demographic and Clinical Characteristics of the Participants

As shown in Table 1, the sample of 554 patients in the study had a mean age of 57.96 years (SD = 11.31; range = 26–85). Among these patients, 73.65% had a bachelor’s or graduate degree, 59.21% were married or partnered, and 64.8% were employed. In terms of financial status, 61.91% had comfortable finance of having more than enough to make ends meet and 31.77% had adequate finance of having enough to make ends meet, but 6.32% had financial hardship of not having enough to make ends meet. While 70.76% of patients were White, 29.24% were non-white: 8.84% (*n* = 49) being Asian, 5.23% (*n* = 29) black or African American, and 15.2% (*n* = 84) Hispanic.

In terms of clinical characteristics (Table 2), 46.93% of the patients had a lumpectomy, 53.07% a mastectomy, 64.08% chemotherapy, 72.2% radiotherapy, and 83.57% hormonal therapy. While 59.21% of the patients underwent axillary lymph node dissection, 40.79% had only sentinel lymph node biopsy. The mean lymph nodes removed was 8.64 (SD = 8.41; range = 1–35). The mean years elapsed since the breast cancer treatment was 4.75 (SD = 5.59; range = 0–43 years). Among the 554 patients, only 3.97% had a history of diabetes, while 22.02% patients had a diagnosis of lymphedema. Between-groups effects sizes for demographic and clinical characteristics are presented in Table 1 and Table 2.

### 3.2. Obesity, Body Fat Percentage and Body Fat Mass

The average BMI of the total patient sample (*n* = 554) was 26.69 (SD = 5.80; range = 16–58.6). Among the 554 patients, 24.36% (*n* = 135) were obese (BMI ≥ 30 kg/m^2^) and 75.63% (*n* = 419) non-obese (BMI < 30 kg/m^2^). The average BMI for patients with obesity was 34.97 (SD = 4.40; range = 30–58.6) while the average BMI for patients with non-obesity was 24.02 (SD = 3.01; range = 16–29.9). Patients with obesity were more likely to be non-white (χ^2^(1) = 12.92, *p* < 0.001; *V* = −0.15), have diabetes (χ^2^(1) = 11.32, *p* < 0.001; *V* = 0.14), have a diagnosis of lymphedema (χ^2^(1) = 7.25, *p* < 0.007; *V* = 0.11), and have financial hardship (χ^2^(1) = 16.52, *p* < 0.001; *V* = 0.17).

As shown in Table 1, the average body fat percentage for the total patient sample (*N* = 554) was 34.71 (*SD* = 8.96; range = 7.8–55.2). Significantly higher body fat percentage (*t* (552) = −20.82; *p* < 0.001; *d* = −2.06) was found in patients with obesity (*M* = 45.17; SD = 4.90; range = 29.2–55.2) compared to patients with non-obesity (*M* = 31.34; SD = 7.20; range = 7.8–51.1). The average body fat mass for the total patient sample (*N* = 554) was 56.52 (*SD* = 25.93; range = 7.5–182.3). Significantly higher body fat mass (*t* (552) = −29.18, *p* < 0.001; *d* = −2.89) was found in patients with obesity (*M* = 92.07; *SD* = 20.44; range = 54.4–182.3) compared to patients with non-obesity *(M =* 45.07; SD = 14.69; range = 7.5–83.7).

### 3.3. The Effect of Obesity on Lymphatic Pain

As shown in Table 3, among the 554 patients, 32.85% experienced lymphatic pain. Significantly more patients with obesity reported lymphatic pain compared to those with non-obesity (51.85% vs. 26.73; χ^2^(1) = 29.21, *p* < 0.001).

As shown in Table 4, to evaluate the effect of obesity on the presence of lymphatic pain given clinical and demographic characteristics as well as body fat percentage, a multivariable logistic regression model fits the data well (McFadden’s Pseudo R^2^ = 0.137; LR χ^2^ (18) = 95.77; *p* < 0.001). Controlling for clinical and demographic characteristics as well as body fat percentage, obesity (B = 1.25; *z* = 3.94; OR = 3.49; 95% CI 1.87–6.50; *p* < 0.001) had the greatest effect on lymphatic pain. Other covariates predicted lymphatic pain, including having financial hardship (B = 1.16; *z* = 2.74; OR = 3.19; 95% CI 1.39–7.31; *p* = 0.006), having radiotherapy (B = 0.58; *z* = 2.13; OR = 1.78; 95% CI 1.05–3.03; *p* = 0.033), and having more lymph nodes removed (B = 0.03; *z* = 2.05; OR = 1.03; 95% CI 1.00–1.06; *p* = 0.041). Subsequent analysis within the obesity group suggests that there is a linear relationship between BMI and lymphatic pain scores amongst patients with a BMI ≥ 30 kg/m^2^ (B = 0.28; 95% CI 0.05–0.41; *t* = 2.50; *p* = 0.014). For obese patients, as BMI increases, levels of lymphatic pain are expected to increase linearly.

### 3.4. The Effect of Obesity on Arm Swelling

As shown in Table 3, among the 554 patients, 39.17% experienced arm swelling. Significantly more patients with obesity reported arm swelling compared to those with non-obesity (57.04% vs. 33.41; χ^2^(1) = 23.92; *p* < 0.001). As shown in Table 5, to evaluate the effects of obesity on the presence of arm swelling given clinical and demographic characteristics as well as body fat percentage, a multivariable logistic regression model fits the data well (McFadden’s Pseudo R^2^ = 0.142; LR χ^2^ (18) = 105.47; *p* < 0.001). Controlling for clinical and demographic characteristics as well as body fat percentage, obesity (B = 1.38; *z* = 4.40; OR = 3.98; 95% CI 1.82–4.43; *p* < 0.001) had the greatest effect on arm swelling. Other significant predictors included having financial hardship (B = 0.89; *z* = 2.03; OR = 2.43; 95% CI 1.03–5.74; *p* = 0.043), having mastectomy (B = 0.78; *z* = 3.13; OR = 2.17; 95% CI 1.40–3.53; *p* = 0.002), having axillary lymph node dissection (B = 0.53; *z* = 2.06; OR = 1.69; 95% CI 1.03–2.80; *p* = 0.040), having radiotherapy (B = 0.80; *z* = 3.03; OR = 2.23; 95% CI 1.33–3.75]; *p* = 0.002), and having more lymph nodes removed (B = 0.03; *z* = 2.05; OR = 1.03; 95% CI 1.00–1.06; *p* = 0.040). Subsequent regression analysis did not reveal any additional effects of BMI on degree of arm swelling among patients with a BMI ≥ 30 kg/m^2^. In other words, patients with a BMI ≥ 30 kg/m^2^ exhibited similar levels of arm swelling, regardless of specific BMI values.

### 3.5. The Effect of Obesity on Truncal Swelling

As shown in Table 3, among the 554 patients, 26.17% experienced truncal swelling. Significantly more patients with obesity reported truncal swelling compared to those with non-obesity (38.52% vs. 22.2; χ^2^(1) = 14.08, *p* < 0.001). As shown in Table 6, to evaluate the effects of obesity on the presence of truncal swelling given clinical and demographic characteristics as well as body fat percentage, a multivariable logistic regression model fits the data well (McFadden’s Pseudo R^2^ = 0.060; LR χ^2^(18) = 38.35; *p* = 0.004). Controlling for clinical and demographic characteristics as well as body fat percentage, obesity (B = 0.61; *z* = 1.95; OR = 1.85; 95% CI 1.00–3.42; *p* = 0.051) had no significant effect on truncal swelling. Significant predictors included having had a mastectomy (B = −0.64; *z* = −2.45; OR = 0.53; 95% CI 0.32–0.88; *p* = 0.014) and younger age (B = −0.02; *z* = −2.10; OR = 0.98; 95% CI 0.96–1.00; *p* = 0.036). In subsequent regression analysis, no additional effects of BMI on truncal swelling were observed among patients with a BMI ≥ 30 kg/m^2^. The results suggest that patients with a BMI ≥ 30 kg/m^2^ are expected to have similar levels of truncal swelling, regardless of specific BMI values.

## 4. Discussion

This is the first study in a larger sample of breast cancer patients that provides initial evidence for quantifiable effects of obesity defined as a BMI ≥ 30 kg/m^2^ on patient-centered health outcomes of lymphatic pain, arm swelling, and truncal swelling. This study extends the previous work that obesity defined as a BMI ≥ 30 kg/m^2^ is a risk factor for objective arm swelling or lymphedema defined as an increased limb size or girth comparing affected (or lymphedematous) and unaffected limbs [19,20] as well as lymphedema defined by >7.1 L-Dex ratio by a Bioelectrical Impedance Analysis (BIA) device [28]. Obesity can influence breast cancer patients’ inflammatory responses, resulting in abnormal lymph fluid accumulation [17,18]. We found that patients with obesity suffered the greatest odds of having lymphatic pain compared to clinical risk factors of having radiotherapy and more lymph nodes removed and demographic risk factor of having financial hardship. Similarly, patients with obesity had the highest odds of having arm swelling compared to clinical factors of having a mastectomy, axillary lymph node dissection, radiotherapy and more lymph nodes removed as well as demographic factor of having financial hardship. Thus, the results support our study hypothesis that breast cancer patients with obesity (BMI ≥ 30 kg/m^2^) would have higher odds of having lymphatic pain and arm swelling than those with non-obesity (BMI < 30 kg/m^2^) controlling for demographic and clinical characteristics as well as body fat percentage. Obesity had no effects on truncal swelling. Furthermore, having a mastectomy and younger age had very small effects on truncal swelling. This suggests that truncal swelling may have different pathological underpinnings. As obesity, lymphatic pain, and arm swelling are inflammatory pathological conditions [4,17,18], future research should explore the inflammatory pathways and understand the molecular mechanisms to find a cure for lymphatic pain and swelling, even for lymphedema.

About one in four patients in our study were obese (BMI ≥ 30 kg/m^2^). Significantly higher body fat percentage and body fat mass were found in patients with obesity compared to patients with non-obesity. Researchers have speculated that body fat percentage and body fat mass may be more accurate measures for obesity than obesity defined as a BMI ≥ 30 kg/m^2^ due to inclusion of lean muscle in BMI [2]. Our results suggest that after controlling for body fat percentage, obesity defined as BMI ≥ 30 kg/m^2^ still had the greatest effects on lymphatic pain and arm swelling. Future research should focus on clinical cutoff points for obesity in terms of body fat percentage or body fat mass in breast cancer patients and compare predictive sensitivity among BMI, body fat percentage, and body fat mass on patient-centered health outcomes.

It should be noted that clinical risk factors (i.e., radiotherapy, mastectomy, more lymph nodes removed, axillary lymph node dissection) and demographic risk factor (i.e., younger age) are non-modifiable risk factors. As the major risk factor for lymphatic pain and arm swelling, obesity is the only risk factor that can be modified through lifestyle changes. This underscores the importance of incorporating obesity management in treating and reducing the risk of lymphatic pain and arm swelling.

Importantly, findings of our study also provide new evidence that patients with financial hardship also had higher odds of having lymphatic pain and arm swelling comparing to the aforementioned non-modifiable clinical and demographic risk factors. In our study, patients with obesity were more likely to be non-white, have diabetes, have lymphedema, and have financial hardship. The finding that having financial hardship results in the increased odds of having lymphatic pain and arm swelling in breast cancer patients is noteworthy. Obesity and financial hardship are strongly intertwined with socioeconomic status, and the interconnection of race and socioeconomic status accounts for substantial racial/ethnic differences in health [29,30]. Taken together, our study suggests that management of obesity and provision of financial support may be beneficial in reducing the risk of developing long-term and debilitating adverse effects of cancer treatment (e.g., lymphatic pain, lymphedema, swelling) and improve the quality of breast cancer survival and decrease years lost due to disability.

### Limitations and Strengths of the study

Limitations of the study include its cross-sectional design that prevents an evaluation of changes in obesity in relation to lymphatic pain, arm swelling, and truncal swelling over time. Nevertheless, this is the first study that provides initial evidence for quantifiable effects of obesity on patient-centered health outcomes of lymphatic pain, arm swelling, and truncal swelling. A strength of the study is the use of a valid and reliable instrument to evaluate symptoms associated with lymph fluid accumulation, which allows for precision classification of lymphatic pain, arm swelling, and truncal swelling. Another strength of the study is to investigate the effects of obesity (BMI ≥ 30 kg/m^2^) after controlling for body fat in relation to muscle mass (i.e., body fat percentage). Finally, a comparatively larger sample permits to quantify the effects of obesity on lymphatic pain, arm swelling, and truncal swelling.

## 5. Conclusions

Obesity, lymphatic pain, arm swelling, and truncal swelling affect a substantial proportion of breast cancer patients. Findings of our study demonstrate that obesity is a major significant risk for lymphatic pain and arm swelling, even after controlling for the effects of demographic and clinical risk factors as well as body fat percentage. As obesity, lymphatic pain, and swelling are inflammatory pathological conditions [4,17,18], future research should explore the inflammatory pathways and understand the molecular mechanisms to find a cure for lymphatic pain and swelling, and even for lymphedema. In addition, obesity is the only risk factor that can be modified through lifestyle changes while financial hardship may be addressed by societal intervention. A holistic approach that incorporates medical, lifestyle, and social support should be considered in treating and reducing the risk of lymphatic pain and arm swelling.

## Figures and Tables

**Figure 1 biomedicines-09-00818-f001:**
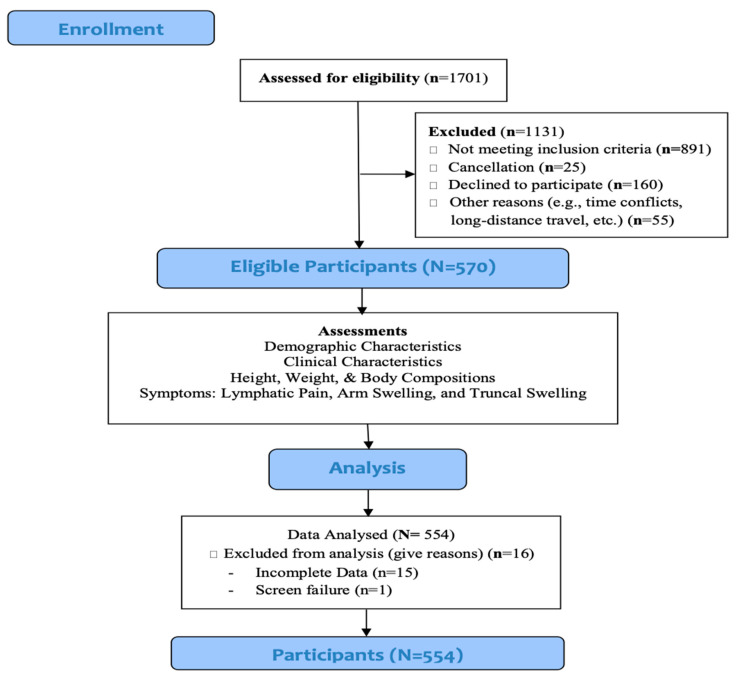
Patient flowchart.

**Table 1 biomedicines-09-00818-t001:** Demographic characteristics of participants (*N* = 554).

	All Sample N = 554	Obesity BMI ≥ 30 kg/m^2^ n = 135	Non-Obesity BMI < 30 kg/m^2^ n = 419	Test Statistics			
Continuous Variables	Mean ± SD^1^ (Range)	Mean ± SD (Range)	Mean ± SD (Range)	t	df ^1^	*p* (*t*-Test)	Effect Size ^2^
**Age (in years)**	57.96 ± 11.31 (26–85)	57.50 ± 11.31 (32–82)	58.11 ± 11.32 (26–85)	0.54	552	0.5916	0.05
**Body fat percentage**	34.71 ± 8.96 (7.8–55.2)	45.17 ± 4.90 (29.2–55.2)	31.34 ± 7.20 (7.8–51.1)	−20.82	552	<0.001	−2.06
**Body fat mass**	56.52 ± 25.93 (7.5–182.3)	92.07 ± 20.44 (54.4–182.3)	45.07 ± 14.69 (7.5–83.7)	−29.18	552	<0.001	−2.89
**Categorical Variables**	**n (%)**	**n (%)**	**n (%)**	**Pearson χ^2^**	**df**	***p* (χ^2^)**	**Effect Size**
**Ethnicity**				12.92	1	<0.001	−0.15
Non-white	162 (29.24)	56 (41.48)	106 (25.30)				
White	392 (70.76)	79 (58.52)	313 (74.70)				
**Marital status**				0.03	1	0.852	−0.01
Married/partnered	328 (59.21)	79 (58.52)	249 (59.43)				
Not married	226 (40.79)	56 (41.48)	170 (40.57)				
**Education**				2.73	2	0.255	0.07
Associate Degree or less	146 (26.35)	41 (30.37)	105 (25.06)				
Bachelor’s Degree	209 (37.73)	53 (39.26)	156 (37.23)				
Graduate Degree	199 (35.92)	41 (30.37)	158 (37.71)				
**Financial status**				16.52	2	<0.001	0.17
Comfortable Finance: Have more than enough to make ends meet	343 (61.91)	68 (50.37)	275 (65.63)				
Adequate Finance: Have enough to make ends meet	176 (31.77)	50 (37.04)	126 (30.07)				
Financial Hardship: Do not have enough to make ends meet	35 (6.32)	17 (12.59)	18 (4.30)				
**Employment status**				0.01	1	0.920	−0.004
Employed	359 (64.80)	87 (64.44)	272 (64.92)				
Unemployed	195 (35.20)	48 (35.56)	147 (35.08)				

^1^ SD: Standard Deviation; df: degree of freedom; ^2^ Effect size: Cohen’s *d* for continuous variables and Cramer’s *V* for categorical variables.

**Table 2 biomedicines-09-00818-t002:** Clinical characteristics of participants (*N* = 554).

	All Sample N = 554	Obesity BMI ≥30 kg/m^2^ n = 135	Non-Obesity BMI < 30 kg/m^2^ n = 419	Test Statistics			
Continuous Variables	Mean ± SD^1^ (Range)	Mean ± SD (Range)	Mean ± SD (Range)	t	df ^1^	*p* (*t*-Test)	Effect Size ^2^
**Years elapsed since the breast cancer treatment**	4.75 ± 5.59 (0–43)	4.37 ± 5.10 (0–27)	4.87 ± 5.72 (0–43)	0.89	552	0.3743	0.09
**Number of lymph nodes removed**	8.64 ± 8.41 (1–35)	8.44 ± 7.44 (1–35)	8.70 ± 8.71 (1–35)	0.32	552	0.7508	0.03
**Categorical Variables**	**n (%)**	**n (%)**	**n (%)**	**Pearson χ^2^**	**df**	***p* (χ^2^)**	**Effect Size**
**Lymphedema diagnosis**				7.25	1	0.007	0.11
Yes	122 (22.02)	41 (30.37)	81 (19.33)				
No	432 (77.98)	94 (69.63)	338 (80.67)				
**Diabetes**				11.32	1	0.001	0.14
Yes	22 (3.97)	12 (8.89)	10 (2.39)				
No	532 (96.03)	123 (91.11)	409 (97.61)				
**Lymph node procedures**				1.50	1	0.221	0.05
Axillary lymph node dissection	328 (59.21)	86 (63.70)	242 (57.76)				
Sentinel lymph node biopsy	226 (40.79)	49 (36.30)	177 (42.24)				
**Radiotherapy**				1.49	1	0.222	0.05
Yes	400 (72.20)	103 (76.30)	297 (70.88)				
No	154 (27.80)	32 (23.70)	122 (29.12)				
**Chemotherapy**				**0.52**	**1**	**0.471**	**0.03**
Yes	355 (64.08)	90 (66.67)	265 (63.25)				
No	199 (35.92)	45 (33.33)	154 (36.75)				
**Hormonal Therapy**				***0.10***	***1***	**0.754**	**0.01**
Yes	463 (83.57)	*114 (84.44)*	*349 (83.29)*				
No	91 (16.43)	*21 (15.56)*	*70 (16.71)*				

^1^ SD: Standard Deviation; df: degree of freedom; ^2^ Effect size: Cohen’s *d* for continuous variables and Cramer’s *V* for categorical variables.

**Table 3 biomedicines-09-00818-t003:** Differences in lymphatic pain, arm swelling and truncal swelling between women with obesity and non-obesity (*N* = 554).

	All Sample N = 554	Obesity BMI ≥30 kg/m^2^ n = 135	Non-Obesity BMI < 30 kg/m^2^ n = 419	Test Statistics		
**Outcome Variables**	**n (%)**	**n (%)**	**n (%)**	**Pearson χ^2^**	**df ^1^**	***p*** **(** **χ^2^** **)**
**Lymphatic Pain**				29.21	1	<0.001
Yes	182 (32.85)	70 (51.85)	112 (26.73)			
No	372 (67.15)	65 (48.15)	307 (73.27)			
**Arm Swelling**				23.92	1	<0.001
Yes	217 (39.17)	77 (57.04)	140 (33.41)			
No	337 (60.83)	58 (42.96)	279 (66.59)			
**Truncal Swelling**				14.08	1	<0.001
Yes	145 (26.17)	52 (38.52)	93 (22.20)			
No	409 (73.83)	83 (61.48)	326 (77.80)			

^1^ df: degree of freedom.

**Table 4 biomedicines-09-00818-t004:** Multivariable logistic regression for lymphatic pain (*N* = 554).

	Multivariable Logistic Regression					
	B ^1^	SE	z-Value	Odds Ratio	OR 95% CI	*p* Value
**Obesity**						
Yes	1.25	0.32	3.94	3.49	(1.87, 6.50)	<0.001
No	--	--	--	--	--	
**Diabetes**						
Yes	0.27	0.49	0.55	1.31	(0.50, 3.43)	0.583
No	--	--	--	--	--	
**Ethnicity**						
Non-white	−0.20	0.23	−0.89	0.82	(0.53, 1.27)	0.374
White	--	--	--	--	--	
**Marital status**						
Married/partnered	−0.15	0.21	−0.70	0.86	(0.57, 1.31)	0.483
Not married	--	--	--	--	--	
**Education**						
Associate Degree or less	0.42	0.26	1.60	1.52	(0.91, 2.54)	0.110
Bachelor’s degree	--	--	--	--	--	
≥Master’s Degree	−0.13	0.24	−0.54	0.88	(0.55, 1.41)	0.592
**Financial status**						
Comfortable finance	0.16	0.22	0.71	1.17	(0.76, 1.81)	0.477
Financial hardship	1.16	0.42	2.74	3.19	(1.39, 7.31)	0.006
Adequate finance	--	--	--	--	--	
**Employment status**						
Employed	0.12	0.23	0.52	1.13	(0.72, 1.79)	0.600
Unemployed	--	--	--	--	--	
**Types of surgery**						
Mastectomy	0.40	0.25	1.60	1.50	(0.91, 2.45)	0.109
Lumpectomy	--	--	--	--	--	
**Lymph node procedures**						
Axillary lymph node dissection	0.42	0.27	1.57	1.52	(0.90, 2.56)	0.117
Sentinel lymph node biopsy	--	--	--	--	--	
**Radiotherapy**						
Yes	0.58	0.27	2.13	1.78	(1.05, 3.03)	0.033
No	--	--	--	--	--	
**Chemotherapy**						
Yes	0.33	0.24	1.36	1.39	(0.87, 2.24)	0.173
No	--	--	--	--	--	
**Hormonal therapy**						
Yes	0.52	0.30	1.74	1.69	(0.94, 3.04)	0.082
No	--	--	--	--	--	
**Age (in years)**	−0.02	0.01	−1.47	0.98	(0.96, 1.01)	0.142
**Years elapsed since the breast cancer treatment**	−0.02	0.02	−0.99	0.98	(0.94, 1.02)	0.323
**Numbers of lymph nodes removed**	0.03	0.02	2.05	1.03	(1.00, 1.06)	0.041
**Body fat percentage**	−0.01	0.02	−0.94	0.99	(0.96, 1.02)	0.348
**Intercept**	−2.29	0.67	−3.40	0.10	(0.03, 0.38)	0.001
**McFadden’s Pseudo R^2^**	0.137					
**LR χ^2^ (18)**	95.77					
**Prob > χ^2^**	<0.001					

^1^ B: Regression Coefficient; SE: Standard Error; CI: Confidence Interval; OR: Odds Ratio; --: Reference Group.

**Table 5 biomedicines-09-00818-t005:** Multivariable logistic regression for arm swelling (*N* = 554).

	Multivariable Logistic Regression					
	B ^1^	SE	z-Value	Odds Ratio	OR 95% CI	*p* Value
**Obesity**						
Yes	1.38	0.31	4.40	3.98	(1.82, 4.43)	<0.001
No	--	--	--	--	--	
**Diabetes**						
Yes	0.20	0.50	0.39	1.22	(0.46, 3.22)	0.695
No	--	--	--	--	--	
**Ethnicity**						
Non-white	−0.09	0.22	−0.39	0.92	(0.60, 1.41)	0.696
White	--	--	--	--	--	
**Marital status**						
Married/partnered	−0.20	0.20	−0.97	0.82	(0.55, 1.22)	0.330
Not married	--	--	--	--	--	
**Education**						
Associate Degree or less	0.35	0.26	1.34	1.41	(0.85, 2.34)	0.179
Bachelor’s degree	--	--	--	--	--	
≥Master’s Degree	−0.12	0.23	−0.51	0.89	(0.56, 1.40)	0.612
**Financial Status**						
Comfortable finance	−0.04	0.22	−0.19	0.96	(0.63, 1.47)	0.848
Financial hardship	0.89	0.44	2.03	2.43	(1.03, 5.74)	0.043
Adequate finance	--	--	--	--	--	
**Employment status**						
Employed	0.002	0.22	0.01	1.00	(0.65, 1.56)	0.992
Unemployed	--	--	--	--	--	
**Types of surgery**						
Mastectomy	0.78	0.25	3.13	2.17	(1.4, 3.53)	0.002
Lumpectomy	--	--	--	--	--	
**Lymph node procedures**						
Axillary lymph node dissection	0.53	0.26	2.06	1.69	(1.03, 2.80)	0.040
Sentinel lymph node biopsy	--	--	--	--	--	
**Radiotherapy**						
Yes	0.80	0.26	3.03	2.23	(1.33, 3.75)	0.002
No	--	--	--	--	--	
**Chemotherapy**						
Yes	0.17	0.23	0.76	1.19	(0.76, 1.87)	0.448
No	--	--	--	--	--	
**Hormonal therapy**						
Yes	−0.02	0.27	−0.06	0.98	(0.58, 1.67)	0.948
No	--	--	--	--	--	
**Age (in years)**	−0.01	0.01	−0.50	0.99	(0.97, 1.02)	0.620
**Years elapsed since the breast cancer treatment**	−0.003	0.02	−0.19	1.00	(0.96, 1.03)	0.852
**Number of lymph nodes removed**	0.03	0.02	2.05	1.03	(1.00, 1.06)	0.040
**Body fat percentage**	−0.03	0.02	−1.93	0.97	(0.94, 1.00)	0.053
**Intercept**	−1.20	0.62	−1.94	0.30	(0.09, 1.01)	0.052
**McFadden’s Pseudo R^2^**	0.142					
**LR χ^2^ (18)**	105.47					
**Prob > χ^2^**	<0.001					

^1^ B: Regression Coefficient; SE: Standard Error; CI: Confidence Interval; OR: Odds Ratio; --: Reference Group.

**Table 6 biomedicines-09-00818-t006:** Multivariable logistic regression for truncal swelling (*N* = 554).

	Multivariable Logistic Regression					
	B ^1^	SE	z-Value	Odds Ratio	OR 95% CI	*p* Value
**Obesity**						
Yes	0.61	0.31	1.95	1.85	(1.00, 3.42)	0.051
No	--	--	--	--	--	
**Diabetes**						
Yes	0.001	0.51	0.00	1.00	(0.37, 2.70)	0.997
No	--	--	--	--	--	
**Ethnicity**						
Non-white	−0.06	0.23	−0.25	0.94	(0.60, 1.48)	0.805
White	--	--	--	--	--	
**Marital status**						
Married/partnered	−0.14	0.21	−0.65	0.87	(0.57, 1.32)	0.514
Not married	--	--	--	--	--	
**Education**						
Associate Degree or less	0.38	0.27	1.43	1.47	(0.87, 2.48)	0.152
Bachelor’s degree	--	--	--	--	--	
≥Master’s Degree	0.26	0.24	1.07	1.30	(0.80, 2.09)	0.286
**Financial status**						
Comfortable finance	0.30	0.22	1.35	1.35	(0.87, 2.09)	0.176
Financial hardship	0.22	0.42	0.52	1.24	(0.55, 2.81)	0.602
Adequate finance	--	--	--	--	--	--
**Employment status**						
Employed	−0.04	0.23	−0.16	0.96	(0.61, 1.52)	0.869
Unemployed	--	--	--	--	--	
**Types of surgery**						
Mastectomy	−0.64	0.26	−2.45	0.53	(0.32, 0.88)	0.014
Lumpectomy	--	--	--	--	--	
**Lymph node procedures**						
Axillary lymph node dissection	0.25	0.26	0.96	1.29	(0.77, 2.17)	0.337
Sentinel lymph node biopsy	--	--	--	--	--	
**Radiotherapy**						
Yes	0.10	0.28	0.36	1.11	(0.64, 1.92)	0.722
No	--	--	--	--	--	
**Chemotherapy**						
Yes	0.13	0.24	0.54	1.13	(0.71, 1.80)	0.591
No	--	--	--	--	--	
**Hormonal therapy**						
Yes	−0.01	0.28	−0.05	0.99	(0.57, 1.71)	0.959
No	--	--	--	--	--	
**Age (in years)**	−0.02	0.01	−2.10	0.98	(0.96, 1.00)	0.036
**Years elapsed since the breast cancer treatment**	−0.04	0.02	−1.85	0.96	(0.91, 1.00)	0.064
**Numbers of lymph nodes removed**	0.004	0.02	0.26	1.00	(0.97, 1.04)	0.793
**Body fat percentage**	0.01	0.02	0.57	1.01	(0.98, 1.04)	0.570
**Intercept**	−1.73	0.67	−2.59	0.18	(0.09, 0.60)	0.010
**McFadden’s Pseudo R^2^:**	0.060					
**LR χ^2^ (18)**	38.35					
**Prob > χ^2^**	0.004					

^1^ B: Regression Coefficient; SE: Standard Error; CI: Confidence Interval; OR: Odds Ratio; --: Reference Group.

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
