# Peer review of "The Effects of Obesity on Lymphatic Pain and Swelling in Breast Cancer Patients"

_biomedicines, 2021, doi:10.3390/biomedicines9070818_

Round 1

Reviewer 1 Report

This paper examines relationship between obesity and lymphatic pain and arm swelling.  The methodology is sound and produces some interesting results.

Having demonstrated categorically that there is a relationship on the basis of obesity with BMI >30, a subsequent analysis by linear regression would be worth doing and should be very easy to calculate (even as a supplemental data if the results are negative) .  For obese people, is the relationship between pain and obesity linear or exponential? Alternatively, a sub-analysis looking at BMI groups say 30-35, 35-40, >40, etc might uncover if the effect is expanded by a greater degree of obesity.  One would predict based on this data that the pain effect is accentuated by morbid obesity, but it is not known with this level of analysis.

Reviewer 2 Report

Authors report a very detailed cross-scectional study regarding the effect of obesity on lymphatic pain and limb swelling in post-surgery breast cancer patients. Data are sound, the employed statistical methods are appropriate and overall the conclusions are well founded on the data reported.

Several suggestions to improve the clarity of the manuscript are presented below:

Lines 57, 64, 65 please remove "lympnatic" and substitute with interstitial

lines 56-57, 63-64 and 66-67 repeat the same concepts. Please leave one instance only of these information.

From line 54 to line 69 refs 14-15 are cited 5 times. Pleas only cite once.

lines 68-69 is there a better way to stage lymphedema severity? the proposed description is very vague.

line 113 and elsewhere: is the use of "fat mass in pounds" really relevant to the reader. If the value is given, please convert it to SI units, otherwise would be better to omit "pounds" and only leave body fat mass.

line 176: the statement here is a repetition of what was written before in lines 137-140. Please remove either one.

Table 1 and table 4: For clarity and readability, lines reporting values of demographic, social, marital status could be extrapolated and put in a separate table, so that only parameters more directly linked with clinical aspect are left together. (i.e. put in a separate table Marital status, education, financial status, employment status).

line 303 maybe would be better to use the term pathophysiollgical, given that truncal swelling is not a physiological phenomenon.

Moreover, to this reviewer it seems that no direct measurement of arm swelling nor staging has been instrumentally performed. This fact might leave some doubts on the reasonable self-assessment of the subjects. This reviewe suggest to state it clearly in the "limitations" of the study. Otherwise, if a clinical assessment (measurement) of the swelling has been performed, please describe it clearly in the methods section.

Reviewer 3 Report

The topic is of great importance from both, an individual and societal perspective. The study is professionally conducted by an expert team, and the report is very well written in all aspects. The set of predictors was wisely selected and operationalised. The obtained associations of the predictors with the outcome (co-occurring pain and arm swelling) are correctly discussed and reasoning does not go beyond the information the data provide.

I have only two minor comments:

Comments:

1) Ad: l. 338-339     “… this is the first study that provides evidence for quantifiable effects of obesity on patient-centered health outcomes of lymphatic pain, arm swelling, and truncal swelling.”

 I would be very careful about the term „The effects of obesity on..“, since:

 1) no data on possible mechanisms are provided (not planned in this study), rather the Author generate new hypothesis (e.g. in Discussion, l. 290-291: “Obesity can influence breast cancer patients’ inflammatory responses, resulting in abnormal lymph fluid accumulation …”, or in Conclusion l.353-355: “future research should explore the inflammatory pathways and understand the molecular mechanisms…”

2) the cross-sectional studies (where risk factors are measured at the same time as the outcome) naturally provide weak evidence for causality and struggle with limited generality,

3) changing the set of predictors changes mutual relationships among predictors, which, in turn, changes regression coefficients, their standard errors, and results of statistical tests (so might change, “the quantifiable effects of obesity on the outcomes”).

What we can obtain with fitting a (multiple) logistic regression model to data, is, whether a predictor in interest brings an information gain in the frame of selected predictors. That means, a significant amount of information for explaining variability in the outcome above that provided by the set of remaining predictors. The authors are discussing odds of having the outcome - a measure of association. The term “effect” usually refers to an estimate from a model, such as a regression.

But the term “effect” has other connotations in other contexts.  Therefore, I would recommend to introduce the meaning of the word "effect" especially regarding the "effects of obesity".

2) I have found a recently issued article on "The Prevalence and Impact of Lymphatic Pain on Daily Living in Breast Cancer Survivors" by Fitzgerald K. (NURSING| VOLUME 22, ISSUE 5, P595-596, MAY 01, 2021) DOI:https://doi.org/10.1016/j.jpain.2021.03.073

The author states:  "Financial instability was a significant predictor of lymphatic pain.

Since "Financial status" was  included  in the model as a predictor, could the authors comment on the published study?

3) unify the math symbols typography
